# The Association of Serum Uric Acid Levels and Various Uric Acid-Related Ratios with Insulin Resistance and Obesity: A Preliminary Study in Adolescents

**DOI:** 10.3390/children10091493

**Published:** 2023-08-31

**Authors:** Okan Dikker, Ayşe Aktaş, Mustafa Şahin, Murat Doğan, Hüseyin Dağ

**Affiliations:** 1Department of Medical Biochemistry, Istanbul Prof. Dr. Cemil Taşcıoğlu City Hospital, University of Health Sciences, Istanbul 34384, Turkey; ayse.aktas25@saglik.gov.tr; 2Department of Medical Biochemistry, Hitit University Faculty of Medicine, Çorum 19030, Turkey; sahinmustafa@hitit.edu.tr; 3Department of Pediatrics, Istanbul Prof. Dr. Cemil Taşcıoğlu City Hospital, University of Health Sciences, Istanbul 34384, Turkey; murat.dogan@argireklam.com (M.D.); huseyin.dag@sbu.edu.tr (H.D.); 4Department of Pediatric Basic Sciences, Institute of Child Health, Adolesance Health, Istanbul University, Istanbul 34093, Turkey

**Keywords:** uric acid, uric acid-to-albumin ratio, uric acid-to-creatinine ratio, uric acid-to-HDL cholesterol ratio, uric acid-to-lymphocyte ratio, obesity, adolescent

## Abstract

Background: Studies have shown that serum uric acid levels and uric acid-related ratios, such as uric acid-to-albumin ratio (UAR), uric acid-to-creatinine ratio (UCR), uric acid-to-high-density lipoprotein cholesterol (HDL cholesterol) ratio (UHR), and uric acid-to-lymphocyte ratio (ULR), are associated with various diseases and their complications, and that these ratios can be used as biomarkers. In the current study, we aimed to investigate uric acid levels in obese adolescents and the relationship of uric acid-related ratios with insulin resistance and obesity for the first time in the literature. Methods: A total of 100 adolescents (60 obese and 40 healthy) aged 10–17 years were retrospectively included. Participants were assigned to two groups: the obese group and the healthy control group. Obesity was defined as a body mass index (BMI) >the 95th percentile for age and gender. Demographic and laboratory data (serum glucose, urea, creatinine, uric acid, albumin, aspartate aminotransferase (AST), alanine aminotransferase (ALT), C-reactive protein (CRP), total cholesterol, triglyceride, HDL cholesterol, thyroid-stimulating hormone (TSH), free T4 (fT4), insulin levels, and complete blood count) were obtained from the laboratory information management system. A homeostatic model of assessment for insulin resistance (HOMA-IR), low-density lipoprotein cholesterol (LDL cholesterol), and uric acid-related ratios were calculated. Results: Uric acid, UAR, UCR, and UHR levels of obese adolescents were significantly higher than the healthy group (*p* < 0.05). We found that HOMA-IR was positively correlated with uric acid, UAR, and UHR. No correlation was found between BMI and uric acid or uric acid-related ratios. We did not find any difference between the two groups in terms of ULR levels, and we did not find any correlation between BMI and HOMA-IR. Conclusion: High levels of serum uric acid, UAR, UCR, and UHR were associated with obesity. Furthermore, we found that uric acid, UAR, and UHR were positively correlated with insulin resistance.

## 1. Introduction

Obesity is a chronic inflammatory disease that remains a major public health problem on a global scale. Obesity is a condition that develops as a result of weight gain when the amount of energy intake per day exceeds the amount spent. It is characterized by an excessive accumulation of lipids in various tissues. Obesity has a multifactorial etiology, including genetic, environmental, socioeconomic, and psychological influences. There is a significantly high incidence of obesity in the pediatric age group. It has been reported that approximately one in every five adolescents is obese and approximately half of them will have obesity in later adulthood [1]. Obesity, a metabolic condition, induces insulin resistance by altering the insulin signaling pathway via a state of mild inflammation. This situation plays a role in the etiopathogenesis of diseases such as type 2 diabetes mellitus, hyperlipidemia, and cardiovascular diseases [2]. Inflammation and insulin resistance are important factors in the deterioration of human health, particularly affecting glucose and lipid metabolism in skeletal muscle, liver, and adipose tissue [3].

Uric acid is the end product of purine metabolism. After uric acid—which also has antioxidant effects—is freely filtered by the renal glomerulus, around 90% is reabsorbed from the renal tubules [4]. Hyperuricemia is a consequence of abnormal purine metabolism, resulting in either an excessive production or abnormal excretion of uric acid [5]. In addition to causing gouty arthritis and nephropathy, hyperuricemia is also related to obesity, insulin resistance, type 2 diabetes, and cardiovascular disease [6]. In some studies, high uric acid levels were reported in obesity [7] and in cases with insulin resistance [8].

A review of the literature showed that several uric acid-related ratios, such as the uric acid/albumin ratio (UAR), uric acid/creatinine ratio (UCR), uric acid/HDL cholesterol ratio (UHR) and uric acid/lymphocyte ratio (ULR), are associated with various diseases and their complications. These parameters can be used as biomarkers with a higher predictive accuracy than uric acid alone. Among these, UAR has been reported to be linked to short-term mortality in patients with acute renal injury [9], the severity of chronic coronary artery disease [10], carotid intima–media thickness in hypertension patients [11] and myocardial infarction [12]. UCR has been reported to be associated with metabolic syndrome [13], the function of beta cells in patients with diabetes mellitus [14], nonalcoholic fatty liver disease (NAFLD) [15], all mortal conditions that may occur due to hypertension [16] and diabetic kidney disease [17]. Moreover, UHR has been reported to be linked to metabolic syndrome [18], visceral body fat in individuals with type 2 diabetes mellitus [19], diabetic control [20], NAFLD risk and fatty liver severity [21], and poorly controlled hypertension [22]. Furthermore, studies have revealed that ULR is associated with the risk of stroke [23], impairment in myocardial contractility [24], and early stage lung cancer [25].

However, no study has demonstrated the correlation between UAR and ULR with BMI and HOMA-IR. Nevertheless, some studies have shown significant correlations between UCR and BMI [13,16] and HOMA-IR [14]. Furthermore, studies have shown that UHR correlates with BMI [19,20] but not with HOMA-IR [19]. The course of uric acid-related ratios in obese adolescents have not been investigated. We have thus planned this study to evaluate the relationship between uric acid-related ratios and insulin resistance and obesity in obese adolescents for the first time in the literature.

## 2. Materials and Methods

### 2.1. Study Design and Population

A total of 100 adolescents were included in our study retrospectively. The data of 40 adolescents aged 10–17 years who admitted to Istanbul Prof Dr. Cemil Taşcıoğlu City Hospital Pediatrics Health and Diseases Outpatient Clinics between January 2022 and January 2023 and were diagnosed with obesity, as well as 40 adolescents who applied for routine health follow-up and had no health problems, were retrieved by the laboratory information management system. Weight and height were measured when participants were wearing light clothing and no shoes. Height was measured using a wall stadiometer with an accuracy of 0.1 cm. Body weight was measured with a digital scale with an accuracy of 0.1 kg. BMI (kg/m^2^) was determined by dividing weight in kg by the square of height in meters. Participants were assigned to 2 groups as obese adolescents and the healthy control group. Obesity was defined as a BMI ≥ 95th percentile for age and gender. The BMI of the healthy control group was normal (BMI ≥ 5th to <85th percentile) according to their age and gender [26]. The ethnicity of the subjects was Turkish. No patient with any reported comorbidity, medication use, or smoking history was included in the study.

### 2.2. Laboratory Analysis

Glucose, urea, creatinine, uric acid, albumin, AST, ALT, CRP, total cholesterol, triglyceride, HDL cholesterol, and LDL cholesterol levels were measured colorimetrically using an autoanalyzer. TSH, free T4, and insulin levels were measured via chemiluminescent immunoassay on the same autoanalyzer (Roche Brand, Cobas 8000 model, San Francisco, CA, USA). HbA1c was measured using high performance liquid chromatography on an autoanalyzer (Bio-Rad, Variant II turbo, Dubai, United Arab Emirates). The complete blood count test was measured on Mindray BC 6800 (Mindray Building, Shenzhen, China). LDL cholesterol was calculated using the Friedewald formula [27]. The HOMA-IR index was calculated as follows: HOMA-IR = glucose (mg/dL) × insulin (IU/L)/405 [28].

### 2.3. Calculations of Uric Acid-Related Ratios

UAR was calculated by dividing uric acid by albumin (mg/dL/mg/dL). The UCR was calculated by dividing uric acid by creatinine (mg/dL/mg/dL). UHR was calculated by dividing uric acid by HDL cholesterol (mg/dL/mg/dL). The ULR was calculated by dividing uric acid by lymphocyte count (mg/dL/10^3^).

### 2.4. Ethical Approval

All subjects presented their informed consent for inclusion before they participated in the study. The study was conducted in accordance with the Declaration of Helsinki, and the protocol was approved by the Ethics Committee of Istanbul Prof. Dr. Cemil Taşcıoğlu City Hospital (Date: 6 March 2023, number: 73).

### 2.5. Statistical Analysis

IBM SPSS Statistics 22 (SPSS Inc., Chicago, IL, USA) programs were used. The suitability of the parameters of the normal distribution was evaluated via the Kolmogorov–Smirnov test. The Student’s *t*-test was used to compare normally distributed parameters between groups. The Mann–Whitney U-test was used for parameters that did not have a normal distribution. Pearson’s correlation analysis was used for normally distributed parameters and Spearman’s Rho correlation analysis was used for non-normally distributed parameters. Chi-squared test and continuity (Yates) correction were used to compare qualitative data. The optimal cut-off points were selected based on receiver operating characteristic (ROC) curve analysis.

## 3. Results

The study was directed with a total of 100 adolescents, 56 girls and 44 boys, aged 10 to 17 years. The average age was 13.7 ± 2.2 years. In terms of mean age and sex distribution, there was no appreciable difference between the groups (*p* > 0.05). In comparison to the healthy control group, obese adolescents’ mean BMI was considerably higher (*p*: 0.001) (Table 1).

Uric acid, UAR, UCR, and UHR were significantly higher in obese adolescents (*p* < 0.05). Serum ULR in obese adolescents was not different from those in controls (*p* > 0.05) (Table 1).

Additionally, the levels of HbA1c, AST, ALT, CRP, triglycerides, insulin, HOMA-IR, white blood cells (WBC), and lymphocytes in obese adolescents were substantially higher than in the controls (*p* < 0.05). However, compared to the control group, the HDL cholesterol and free T4 levels of obese adolescents were considerably lower (*p* < 0.05) (Table 1).

Positive correlations were identified between uric acid, UAR, and HOMA-IR; negative correlations were established between UCR, age, and mean platelet volume (MPV); and positive correlations between lymphocyte count and uric acid were found. Furthermore, UHR was negatively correlated with cholesterol, whereas HOMA-IR was positively correlated with cholesterol. In addition, there was a remarkable negative correlation between ULR and cholesterol (*p* < 0.05) (Table 2).

ROC curves were generated in the diagnosis of obesity.

**Determination of cut-off point for uric acid:** The area under the curve was 0.712. The standard error was 0.06 (*p*: 0.004). The cut-off point for uric acid level in the diagnosis of obesity was >4.4. The sensitivity: 73.3%; specificity: 70% (Figure 1).

**Determination of cut-off point for UAR:** The area under the curve was 0.738. The standard error was 0.05 (*p*: 0.001). The cut-off point determined for the UAR level in the diagnosis of obesity was >1.02. The sensitivity: 70%; specificity: 75% (Figure 2).

**Determination of cut-off point for UCR:** The area under the curve was 0.697. The standard error was 0.05 (*p*: 0.001). The cut-off point determined for the UCR level in the diagnosis of obesity was >6.94. The sensitivity: 85%; specificity: 50% (Figure 3).

**Determination of cut-off point for UHR:** The area under the curve was 0.746. The standard error was 0.05 (*p*: 0.001). The cut-off point determined for the UHR level in the diagnosis of obesity was >0.08. The sensitivity: 80%; specificity: 65% (Figure 4).

## 4. Discussion

### 4.1. Assessment of High Uric Acid Levels

In this study, we found that obese adolescents had high uric acid levels. Studies in the literature have reported high uric acid levels in obesity in both adolescents [7,29] and adults [30]. Studies have demonstrated that obesity increases xanthine oxidase activity in adipose tissue, leading to a higher uric acid production and lower renal clearance of uric acid. The prevalence of hyperuricemia is several times higher in adolescents with obesity [7]. It has been reported that high insulin levels and insulin resistance may be associated with the emergence and development of hyperuricemia in obese individuals. It has been stated that individuals with obesity and hyperuricemia require structured weight loss programs, an evaluation of glucose and lipid metabolism disorders, as well as an improvement of hyperinsulinemia and insulin resistance [31]. Miranda et al. [29] revealed a positive correlation between uric acid levels and HOMA-IR in obese subjects. Gil-Campos et al. [32] found that prepubertal obese children had elevated uric acid levels and that their uric acid levels were positively correlated with BMI and HOMA-IR. However, in the current study, we did not find any correlation between uric acid levels and BMI. Yet, we found a positive correlation between uric acid and HOMA-IR in obese adolescents with high uric acid levels. Thus, uric acid appears to be associated with obesity and insulin resistance. There is no clear understanding of the mechanism underlying insulin resistance in obese children. Therefore, to clarify the mechanism underlying the association between high uric acid, insulin resistance, and obesity, further research is required. Moreover, in our study, we found in the ROC analysis that a uric acid level of >4.4 mg/dL could predict obesity in the adolescent group with a sensitivity of 73.3% and a specificity of 70%. Hence, we consider that uric acid could be used as an indicator of obesity in adolescents.

### 4.2. Assessment of High UAR

The UAR, derived from serum uric acid and albumin, has been reported in studies as a novel inflammatory marker. It has been reported that the UAR may have a better predictive performance in certain diseases or their complications compared to its components. Özgür et al. reported that the UAR has been shown to be a strong predictor of short-term mortality in patients with acute renal injury [9]. Yeter et al. reported that a UAR higher than 1.7 was significantly associated with acute kidney injury and long-term mortality [33]. They showed that UAR may be a predictor of mortality and acute kidney injury. Yalçınkaya et al. suggested that the UAR predicted disease burden in chronic coronary artery disease [10]. They concluded that it may prove to be a useful biomarker in selecting patients for further evaluation. Şaylık et al. revealed that UAR could be used to predict high carotid intima–media thickness and might be helpful for risk stratification in hypertensive patients [11]. Çınar et al. found that the UAR was high in no-reflow in ST elevation, and they documented that UAR has a predictive value in these patients [12]. Selçuk et al. found that patients with ST-elevated myocardial infarction with new-onset atrial fibrillation had a high UAR [34]. They demonstrated UAR to be an independent predictor of these patients. These results suggest that UAR may be a useful marker that can be used in different ways to diagnose a variety of diseases.

In addition to serum levels of uric acid, which we found to be high in our study, serum albumin levels were statistically similar to those of healthy controls. However, we found that UAR was higher in obese adolescents. We also found that UAR was positively correlated with HOMA-IR, similar to uric acid. Unfortunately, it did not have any correlation with BMI. Studies examining UAR as markers for disease are rare in the literature. Because none of these studies included a correlation analysis of this ratio with BMI or HOMA-IR, it was not possible to directly compare our results with those of these studies. The mechanisms by which the UAR participates in obesity and insulin resistance remain unclear. Furthermore, we found that the optimum UAR value in predicting obesity in adolescents was >1.02 with a sensitivity of 70% and a specificity of 75%. Based on the current study, we suggest that high UAR can be used as a simple and readily available marker of obesity. We believe that in future studies, high UAR can select patients for the further evaluation of obesity and obesity-related complications such as insulin resistance, fatty liver, hypertension, and cardiovascular diseases.

### 4.3. Assessment of High UCR

Kidney plays a key role in the excretion of uric acid. Creatinine, a chemical waste product of creatine, is affected by the number of muscles, food intake, and kidney function. The UCR represents the metabolic status, excluding the influence of kidney function [35]. The UCR has emerged as a new biomarker and is considered an excellent indicator of uric acid production [16]. In the current study, we found that the UCR was higher in obese adolescents. However, the UCR did not show any correlation with BMI and HOMA-IR. Tao et al. found that the UCR was high in metabolic syndrome and that this ratio was associated with the risk of metabolic syndrome [13]. They reported a positive correlation between UCR and BMI. Chen et al. stated that UCR is an independent risk factor for diabetic kidney disease in patients with diabetes mellitus and may help in defining non-albuminuric diabetic kidney disease [17]. They found a positive correlation between UCR and BMI. Ma et al. found that the UCR was positively correlated with the incidence of NAFLD, and the UCR was positively correlated with BMI and HOMA-IR [36]. Li et al. concluded that the UCR correlates with β-cell function in type 2 diabetes mellitus and found a positive correlation between UCR and HOMA-IR in their study [14]. Kawamoto et al. reported that baseline UCR was independently and significantly associated with mortality in hypertensive patients [16]. They reported that UCR correlated with BMI in females, but not in males. Al-Daghri et al. reported that UCR in patients with type 2 diabetes mellitus is strongly associated with the risk of metabolic syndrome, and this ratio can be used as a marker for metabolic syndrome [37]. While they found a correlation between this ratio and BMI in their study, they did not detect any correlation with HOMA-IR. Based on the data of the studies, the correlations of UCR with BMI and HOMA-IR are confounding. In our ROC analysis, we calculated that the optimum UCR value in detecting obesity in adolescents was >6.94 with a sensitivity of 85% and a specificity of 50%. In light of our study data, we consider that high UCR could be useful in predicting obesity in adolescents. Nonetheless, further studies are needed to elucidate its relationship to obesity and its complications.

### 4.4. Assessment of High UHR

Low HDL cholesterol is as well characterized as one of the features of obesity [38]. Likewise, we found lower HDL cholesterol levels in obese adolescents in our study. The UHR, combined with these two metabolic parameters, is a stronger marker of metabolic deterioration [22]. In the literature, only two studies reported a correlation between UHR and BMI [19,20] and one study reported no correlation with HOMA-IR [19]. Among them, Sun et al. reported that the UHR was positively correlated with the visceral-fat-area type-2-diabetes patients and may be a useful and appropriate additional marker for metabolic risks [19]. Aktaş et al. revealed that UHR increased in poorly controlled type-2-diabetes patients [20]. They reported that the UHR could serve as a promising predictor of diabetic control in men with type 2 diabetes mellitus. Yazdi et al. suggested that the UHR increases metabolic syndrome, can help in diagnosing metabolic syndrome, and can be used to screen people with metabolic syndrome risk [18]. Xie et al. reported that an increased UHR was independently associated with an increased risk of NAFLD and a severity of fatty liver [21]. Some studies report that the evaluation of UHR may be helpful in patients with hypertension, as high UHR levels are correlated with poor blood pressure control [22]. Indeed, in another study, the ratio of UHR was found to be at high levels in Hashimoto’s thyroiditis. It has been reported that the ratio of UHR is a useful marker for Hashimoto’s thyroiditis [39]. It has been shown in the literature that UHR may be useful as a marker in a variety of conditions.

In the current study, we showed that UHR was higher in obese adolescents. We also found that UHR was positively correlated with HOMA-IR, similarly to uric acid and UAR. However, it did not have any correlation with BMI. In our study, we found that the optimum UHR value in the diagnosis of obesity in adolescents was >0.08 with a sensitivity of 80% and a specificity of 65%. According to our study data, we believe that high UAR can be used as a marker of obesity in adolescents, that high UHR may be associated with obesity and its complications, and may be a useful marker in the selection of patients for further evaluation.

### 4.5. Assessment of ULR

It is well-documented that the lymphocyte count is high in obese children [40]. Similarly, in our study, we found higher lymphocyte counts in obese adolescents. The ULR shows a joint effect of uric acid and lymphocyte count. There are few studies on ULR in the literature, and none of these studies found a relationship between BMI and HOMA-IR. Tian et al. reported that ULR was associated with the risk of hemorrhagic stroke, but not with ischemic stroke [23]. Günay et al. suggested that the levels of ULR at the time of presentation in ST-elevated myocardial infarction patients could be used as an independent indicator of the deterioration in myocardial contraction performance, and could be used as an inexpensive and easily accessible routine laboratory parameter [24]. Yang et al. reported that ULR could be considered a strong risk stratification marker to refine prognostic prediction for early stage lung cancer [25]. Wei et al. reported that the ULR produced more prognostic values in elders with rheumatic heart disease undergoing valve replacement surgery, which can be considered as a preoperative risk-stratified method [41]. In our study, we found that the ULR in obese adolescents did not differ significantly in obese adolescents and did not have any correlation with BMI and HOMA-IR. Our findings suggest that ULR is not associated with obesity and insulin resistance. Nevertheless, further studies are needed to clarify this finding. 

As in the current study, high serum uric acid levels have long been associated with obesity. Furthermore, this study demonstrated that various serum uric acid-related ratios were associated with obesity in adolescent. The study findings suggest the following: (1) our study will contribute to the methodology of future studies, as uric acid-related ratios may be found at different levels in obese patients and (2) in addition, when these ratios are given alongside the routine test results, it may be a warning to clinicians in terms of obesity, insulin resistance, and related complications, especially for patients in regions where examination conditions are limited.

### 4.6. Limitations

There are a few limitations to our study, including its retrospective design, which may make it difficult to interpret the results, and its relatively small sample size. We did not investigate the correlations with complications of obesity and, since all subjects were Turkish people, the utility of the uric acid-related ratios should be checked in different ethnical populations. Thus, more large prospective clinical studies are needed to confirm our findings. 

## 5. Conclusions

High levels of uric acid, UAR, UCR, and UHR are associated with obesity. Furthermore, we found that uric acid, UAR, and UHR were positively correlated with insulin resistance. 

## Figures and Tables

**Figure 1 children-10-01493-f001:**
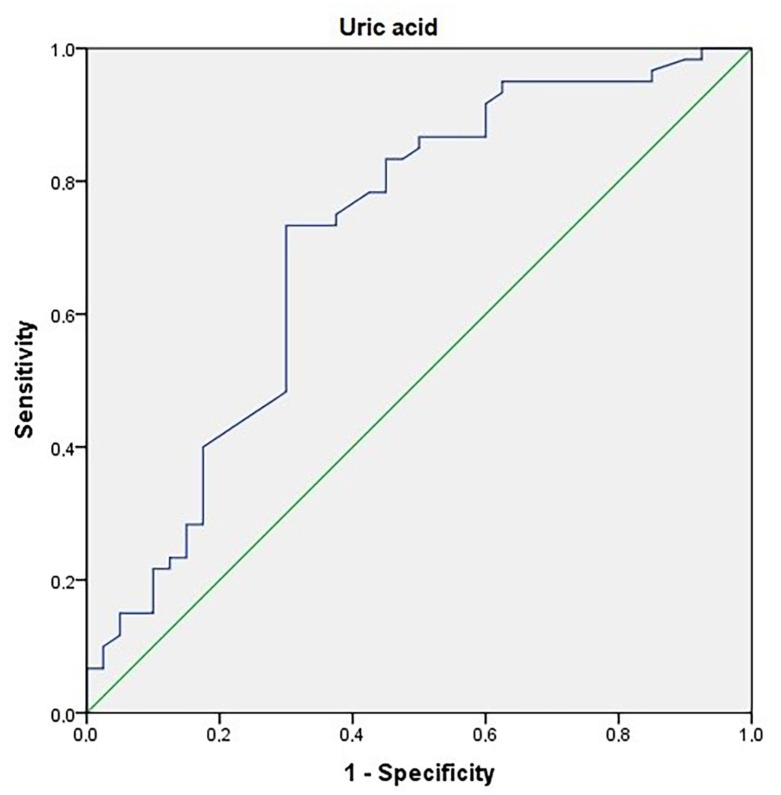
An ROC curve for serum uric acid levels in the prediction of obesity.

**Figure 2 children-10-01493-f002:**
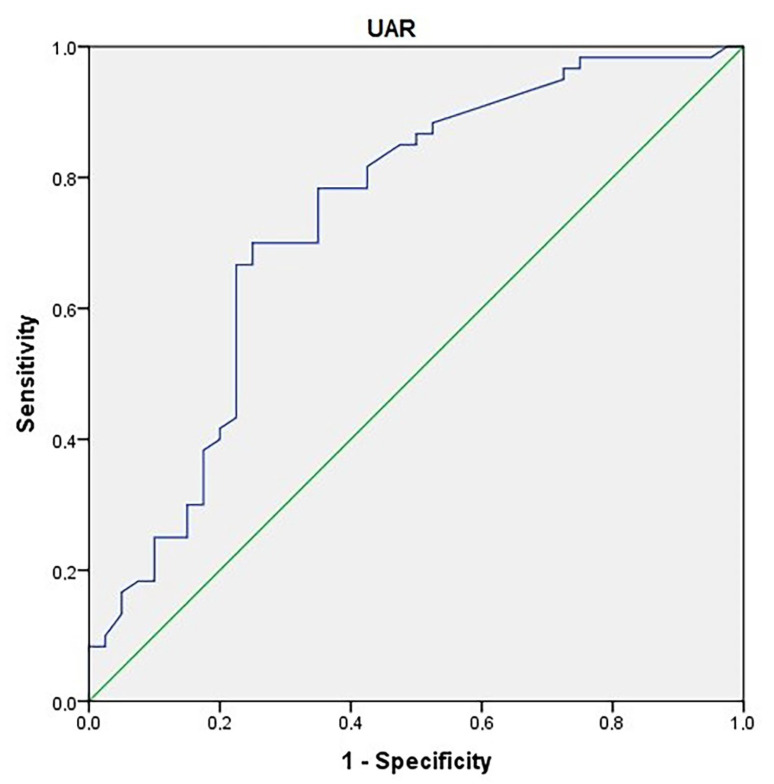
An ROC curve for uric acid-to-albumin ratio in the prediction of obesity.

**Figure 3 children-10-01493-f003:**
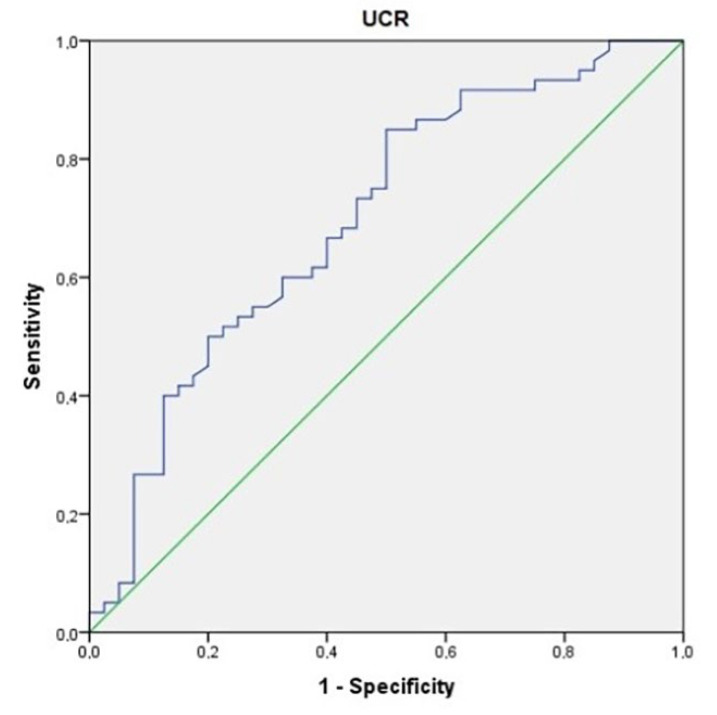
An ROC curve for uric acid-to-creatinine ratio in the prediction of obesity.

**Figure 4 children-10-01493-f004:**
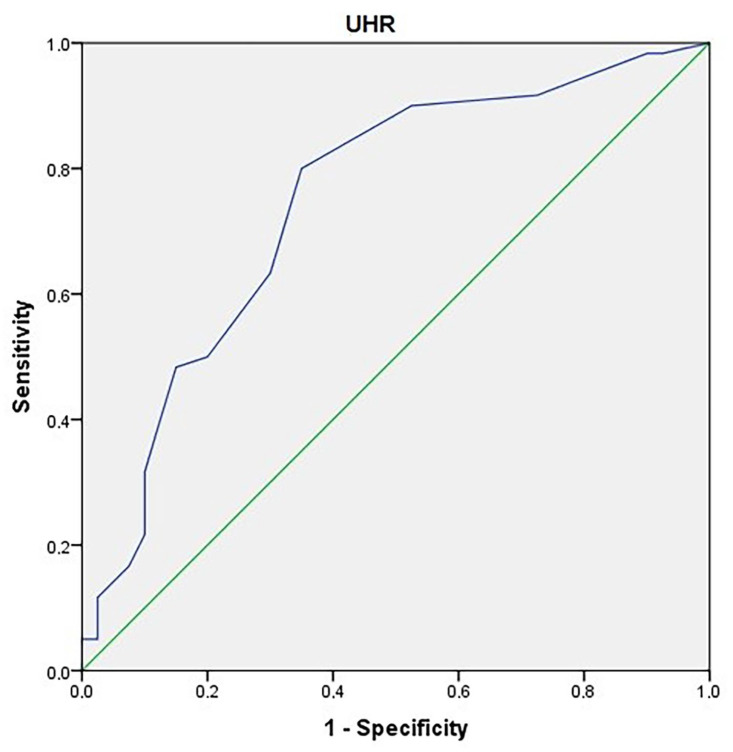
An ROC curve for uric acid-to-HDL cholesterol ratio in the prediction of obesity.

**Table 1 children-10-01493-t001:** Demographic and laboratory data on obese adolescents and healthy controls.

	Obese Adolescents (n: 60)	Healthy Control (n: 40)	
	Mean ± SD (Median)	Mean ± SD (Median)	*p*
** * Demographic data * **			
Age _Mean ± SD_	13.62 ± 2.15	14.02 ± 2.29	0.367 ^1^
**BMI _Mean ± SD_**	**31.66 ± 4.58**	**20.70 ± 2.40**	**0.001** ^1^
Gender _n (%)_			
Female	26 (%65)	30 (%50)	0.202 ^3^
Male	14 (%35)	30 (%50)	
** * Serum uric acid and uric acid-related ratios * **			
**Uric acid (mg/dL)**	**5.19 ± 1.3 (4.9)**	**4.32 ± 1.12 (3.9)**	**0.001** ^2^
**UAR**	**1.16 ± 0.28 (1.1)**	**0.94 ± 0.24 (0.9)**	**0.001** ^2^
**UCR**	**9.36 ± 2.53 (9.4)**	**7.86 ± 2.45 (7.1)**	**0.001** ^2^
**UHR**	**0.12 ± 0.04 (0.1)**	**0.09 ± 0.03 (0.1)**	**0.001** ^2^
ULR	1.99 ± 0.76 (1.9)	1.82 ± 0.58 (1.7)	0.218 ^2^
** * Other laboratory data * **			
**HbA1c (%)**	**5.42 ± 0.31 (5.4)**	**5.04 ± 0.43 (5)**	**0.001** ^1^
Glucose (mg/dL)	87.98 ± 8.48 (87)	88.33 ± 8.44 (88)	0.844 ^1^
Urea (mg/dL)	22.72 ± 6.62 (21.5)	21.53 ± 6.01 (21)	0.432 ^2^
Creatinine (mg/dL)	0.57 ± 0.12 (0.5)	0.58 ± 0.16 (0.6)	0.740 ^1^
Albumin (mg/dL)	4.48 ± 0.24 (4.5)	4.59 ± 0.39 (4.7)	0.070 ^1^
**AST (U/L)**	**22.5 ± 10.46 (20)**	**17.43 ± 4.75 (16.5)**	**0.003** ^2^
**ALT (U/L)**	**23.52 ± 18.03 (18.5)**	**13.85 ± 7.47 (12)**	**0.001** ^2^
**CRP (mg/L)**	**4.88 ± 3.91 (4.3)**	**1.52 ± 1.88 (0.5)**	**0.001** ^2^
Cholesterol (mg/dL)	156.98 ± 31.87 (150.5)	149.63 ± 27.64 (150)	0.236 ^1^
**Triglyceride (mg/L)**	**109.98 ± 50.59 (96)**	**77.5 ± 29.01 (72)**	**0.001** ^2^
**HDL cholesterol (mg/dL)**	**47.45 ± 10.04 (47)**	**54.65 ± 12.25 (53.5)**	**0.002** ^2^
LDL cholesterol (mg/dL)	88.03 ± 24.84 (86.5)	80.18 ± 24.67 (76)	0.095 ^2^
**TSH (mU/L)**	**2.48 ± 1.23 (2.2)**	**1.98 ± 1.02 (1.9)**	**0.034** ^1^
**Free T4 (ng/L)**	**9.76 ± 1.82 (9.4)**	**11.77 ± 2.17 (11.8)**	**0.001** ^1^
**Insulin (IU/L)**	**18.39 ± 11.95 (15.8)**	**10.21 ± 4.4 (9.7)**	**0.001** ^1^
**HOMA-IR**	**4.09 ± 3.14 (3.3)**	**2.26 ± 1.07 (2.1)**	**0.001** ^2^
**WBC (10^3^)**	**8.05 ± 1.51 (8.1)**	**7.33 ± 1.72 (7)**	**0.029** ^1^
Neutrophil count (10^3^)	4.56 ± 1.39 (4.3)	4.1 ± 1.37 (3.8)	0.077 ^2^
**Lymphocyte count (10^3^)**	**2.78 ± 0.68 (2.8)**	**2.47 ± 0.61 (2.4)**	**0.027** ^1^
RBC (10^6^)	4.88 ± 0.38 (4.9)	4.86 ± 0.46 (4.8)	0.814 ^1^
Platelet count (10^3^)	329.45 ± 73.54 (329.5)	313.13 ± 69.72 (308.5)	0.270 ^1^
MPV (fL)	9.52 ± 1.08 (9.5)	9.78 ± 0.95 (9.8)	0.224 ^1^

^1^ Student’s *t*-test; ^2^ Mann–Whitney U Test; ^3^ chi-squared test and continuity (Yates) correction. The values in bold show statistically significant differences (*p* < 0.05). ALT: Alanine aminotransferase. AST: Aspartate aminotransferase. BMI: Body mass index. CRP: C-reactive protein. HOMA-IR: Homeostatic model of assessment for insulin resistance. HDL cholesterol: High-density lipoprotein cholesterol. LDL cholesterol: Low-density lipoprotein cholesterol. MPV: Mean platelet volume. RBC: Red blood cell. SD: Standard deviation. TSH: Thyroid-stimulating hormone. UAR: Uric acid-to-albumin ratio. UCR: Uric acid-to-creatinine ratio. UHR: Uric acid-to-HDL cholesterol ratio. and ULR: Uric acid-to-lymphocyte ratio. WBC: White blood cell.

**Table 2 children-10-01493-t002:** Correlations in the obese adolescent group (n: 60).

		Uric Acid	UAR	UCR	UHR	ULR
**^1^ Age**	**r**	0.023	0.023	−0.416	0.032	0.229
	**p**	0.863	0.862	0.001	0.806	0.079
**^1^ BMI**	**r**	0.196	0.222	−0.049	0.099	0.122
	**p**	0.133	0.088	0.710	0.451	0.353
**^1^ HbA1c**	**r**	−0.214	−0.142	−0.134	−0.185	−0.074
	**p**	0.100	0.278	0.307	0.157	0.574
**^1^ Glucose**	**r**	0.185	0.132	0.046	0.213	0.001
	**p**	0.157	0.314	0.727	0.102	0.991
**^1^ Albumin**	**r**	0.218	-	0.085	0.222	0.089
	**p**	0.095	-	0.517	0.088	0.499
**^2^ CRP**	**r**	0.052	0.083	0.251	0.068	−0.017
	**p**	0.693	0.527	0.053	0.605	0.899
**^1^ Cholesterol**	**r**	−0.158	−0.102	−0.032	**−0.346**	**−0.263**
	**p**	0.229	0.439	0.811	**0.007**	**0.042**
**^2^ Triglyceride**	**r**	0.091	0.101	0.118	0.180	−0.063
	**p**	0.491	0.442	0.370	0.169	0.631
**^2^ HDL cholesterol**	**r**	−0.168	−0.151	−0.192	-	−0.070
	**p**	0.198	0.250	0.141	-	0.595
**^2^ LDL cholesterol**	**r**	−0.004	0.066	0.112	−0.131	−0.188
	**p**	0.975	0.616	0.394	0.319	0.150
**^1^ Insulin**	**r**	0.195	0.170	0.000	0.023	0.082
	**p**	0.135	0.195	1.000	0.859	0.531
**^2^ HOMA-IR**	**r**	**0.274**	**0.296**	0.126	**0.327**	0.040
	**p**	**0.006**	**0.003**	0.213	**0.001**	0.692
**^1^ Lymphocyte count**	**r**	0.205	0.176	**0.360**	0.159	-
	**p**	0.116	0.178	**0.005**	0.225	-
**^1^ MPV**	**r**	−0.112	−0.132	**−0.321**	0.001	0.075
	**p**	0.394	0.313	**0.012**	0.991	0.571

^1^ Pearson correlation analysis; ^2^ Spearman’s rho correlation analysis. The values in bold show statistically significant differences (*p* < 0.05).

## Data Availability

Data are contained within this article (Table 1 and Table 2, Figure 1, Figure 2, Figure 3 and Figure 4).

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
