# Peer review of "The Association of Serum Uric Acid Levels and Various Uric Acid-Related Ratios with Insulin Resistance and Obesity: A Preliminary Study in Adolescents"

_children, 2023, doi:10.3390/children10091493_

Round 1

Reviewer 1 Report

General comment

The paper presents an effort to describe the relationship between the uric acid level / uric acid-related ratios and obesity. The topic of the manuscript is of high importance, since the prevalence of obesity is increasing not only in adults, but also in children.

I recommend accepting this paper for publication in Children after a minor revision, because:

1) The main purpose of the analysis is not clear. To predict obesity by using uric acid level or its ratios? BMI and the cut-offs and overweightness are much better predictors than any metabolic marker that accompanies obesity. The main purpose of the study should be revised. A suggestion: only to study the relationship between nutritional status (obesity) and uric acid and uric acid related ratios in adolescence?

2) The followings should be checked in the tables and the text: all the abbreviations should be explained before the first use, the values in bold in the tables – I guess – signs significant differences, correlations, … - this should be mentioned in the title or legend of the tables,

3) English should be revised (I know, my English is not good enough to criticize the language of manuscripts written in English, but in this case the title also needs revision, so that is why I took the risk of this suggestion).

I list my specific/minor comments and suggestions to the manuscript in the order of the chapters of the manuscript:

Abstract

A1: “The results were compared statistically.” – there is no information in this sentence, please complete or delete it.

A2: Please give the explanation for the abbreviation HOMA-IR as it is used the very first time in the manuscript.

A3: “Based on this information, we intended to explore how uric acid levels change in obese adolescents and, for the first time in the literature, to examine the relationship between uric acid-related ratios and insulin resistance and obesity to determine if they can serve as useful markers.” – it is not clear from this sentence the Authors search biomarkers for what? For obesity? For insulin resistance?

Introduction

I1: “Based on this information, we intended to explore how uric acid levels change in obese adolescents …” – not the change of urid acid level was studied in obese patients, only the tendency between obese and non-obese adolescents, please correct the sentence.

Material and Methods

M1: “Obesity was defined as a body mass index (BMI) of ≥ 95th percentile for age and gender.” – which references were used? Please name the CDC 2002 references. Why were these references chosen for this analysis? There are not Turkish BMI references or updated/more adequate references than references from the early 2000?

Tables

T1: I guess that Cender means Gender in Table 1, please correct it.

T2: I guess chi2 test was used to check the homogeneity by grouping, “2 Continuity (yates) düzeltmesi” is not clear for me in the respect, please correct the information on the homogeneity test in Table 1.

T3: Student t test was used for comparing BMI of obese and healthy subgroups, BMI is not normally distributed, please check and repeat the analysis (Table 1).

T4: What does ‘-‘ sign mean in Table 4? Not-significant correlation or missing analysis?

Conclusions

C1: “Routine use of uric acid and some uric acid-related ratios may be useful in predicting obesity.” BMI and its cut-offs are the best and easiest tool for checking nutritional status. If the uric acid and uric acid-related ratios are not normal obesity has existed for a long time.

English language of the manuscript should be revised.

Author Response

Dear Editor and Dear Reviewer; First of all thank you very much for your efforts and your comments on our manuscript. We have changed our study based on the reviewer comments. 

'Responses to the 1st reviewer' file is attached

Reviewer 2 Report

Manuscript ID: children-2563720 Authors: Okan Dikker et al.

Title: "The association of serum uric acid levels and various uric acid-related ratios with insulin resistance and obesity: A preliminer study in adolescent"

Journal: Children

The authors of the present study tried to investigate the associations between uric acid levels and related ratios (namely, uric acid to albumin ratio (UAR), uric acid to creatinine ratio (UCR), uric acid to HDL cholesterol ratio (UHR), and uric acid to lymphocyte ratio (ULR)) with insulin resistance and obesity, utilizing data from a cohort of 100 adolescents. The levels of uric acid, UAR, UCR, and UHR were significantly higher in the obese group compared to the normal-weight group. Additionally, a positive correlation was identified between HOMA-IR and uric acid, UAR, and UHR. The following points merit consideration.

 Comments:

1.     Please clarify whether the glucose and insulin values used in the HOMA-IR equation were obtained from fasting measurements. Moreover, add the relevant references for the Friedewald formula and HOMA-IR equation.

2.     In the statistical analysis, please define the method by which the assessment of normal distribution was conducted. Furthermore, clarify how categorical variables were assessed.

3.     Tables 1, 2, and 3 could be merged.

4.     In the discussion section, it would be beneficial to describe the clinical importance and the potential implications arising from the findings of the current study.

5.     Did the authors consider conducting additional analyses (e.g., regression analysis incorporating adjustments for potential covariates, mediation analysis, etc.) to further explore the study's hypothesis?

6.     Please check the manuscript for typos and non-English characters.

Author Response

Dear Editor and Dear Reviewer; First of all thank you very much for your efforts and your comments on our manuscript. We have changed our study based on the reviewer comments.

'Responses to the 2nd reviewer' file is attached

Round 2

Reviewer 2 Report

Manuscript ID: children-2563720 (Revised version) Authors: Okan Dikker et al.

Title: "The association of serum uric acid levels and various uric acid-related ratios with insulin resistance and obesity: A preliminary study in adolescent"

Journal: Children

The authors have satisfactorily responded to my comments and made the necessary changes to the manuscript. There are no further considerations.